# Real-World Effectiveness of the mRNA COVID-19 Vaccines in Japan: A Case–Control Study

**DOI:** 10.3390/vaccines10050779

**Published:** 2022-05-14

**Authors:** Megumi Hara, Takeki Furue, Mami Fukuoka, Kentaro Iwanaga, Eijo Matsuishi, Toru Miike, Yuichiro Sakamoto, Naoko Mukai, Yuki Kinugasa, Mutsumi Shigyo, Noriko Sonoda, Masato Tanaka, Yasuko Arase, Yosuke Tanaka, Hitoshi Nakashima, Shin Irie, Yoshio Hirota

**Affiliations:** 1Department of Preventive Medicine, Faculty of Medicine, Saga University, 5-1-1 Nabeshima, Saga 849-8501, Japan; 2Fukuoka City Office, 1-8-1 Tenjin, Chuohku, Fukuoka 810-8620, Japan; furue.t01@city.fukuoka.lg.jp (T.F.); tanaka.m67@city.fukuoka.lg.jp (M.T.); arase.y01@city.fukuoka.lg.jp (Y.A.); 3Saga-Ken Medical Centre Koseikan, 400 Kase-Town, Saga 840-8571, Japan; fukuoka-m@koseikan.jp (M.F.); iwanaga-k@koseikan.jp (K.I.); matsuishi-e@kosiekan.jp (E.M.); 4Department of Emergency Medicine, Saga University Hospital, 5-1-1 Nabeshima, Saga 849-8501, Japan; miike@cc.saga-u.ac.jp (T.M.); sakamoy@cc.saga-u.ac.jp (Y.S.); 5Fukuoka City Jonan Ward Public Health Center, 6-1-1 Torikai, Jonan Ward, Fukuoka 814-0192, Japan; mukai.n02@city.fukuoka.lg.jp; 6Fukuoka City Higashi Ward Public Health Center, 2-54-1 Hakozaki, Higashi Ward, Fukuoka 812-8653, Japan; kinugasa.y01@city.fukuoka.lg.jp; 7Fukuoka City Hakata Ward Public Health Center, 2-9-3 Hakataeki-Mae, Hakata Ward, Fukuoka 812-8512, Japan; shigyo.m01@city.fukuoka.lg.jp; 8Fukuoka City Sawara Ward Public Health Center, 2-1-1 Momochi, Sawara Ward, Fukuoka 814-8501, Japan; sonoda.n01@city.fukuoka.lg.jp; 9Kanenokuma Hospital, SOUSEIKAI Medical Group (Medical Co. LTA), Hakata-Ku, Fukuoka 812-0863, Japan; yosuke-tanaka@lta-med.or.jp; 10SOUSEIKAI Medical Group (Medical Co. LTA), Hakata-Ku, Fukuoka 812-0025, Japan; hitoshi-nakashima@lta-med.com (H.N.); shin-irie@lta-med.com (S.I.); 11Clinical Epidemiology Research Center, SOUSEIKAI Medical Group (Medical Co. LTA), Higashi-Ku, Fukuoka 813-0017, Japan; hiro8yoshi@lta-med.com

**Keywords:** COVID-19 vaccine, mRNA vaccine, vaccine effectiveness, preventive health behavior

## Abstract

The real-world effectiveness of the coronavirus disease 2019 (COVID-19) vaccines in Japan remains unclear. This case–control study evaluated the vaccine effectiveness (VE) of two doses of mRNA vaccine, BNT162b2 or mRNA-1273, against the delta (B.1.617.2) variant in the Japanese general population in the period June–September 2021. Individuals in close contact with COVID-19 patients were tested using polymerase chain reaction (PCR). A self-administered questionnaire evaluated vaccination status, demographic data, underlying medical conditions, lifestyle, personal protective health behaviors, and living environment. Two vaccine doses were reported by 11.6% of cases (*n* = 389) and 35.2% of controls (*n* = 179). Compared with controls, cases were younger and had a lower proportion who always performed handwashing for ≥20 s, a higher proportion of alcohol consumers, and a lower proportion of individuals living in single-family homes or with commuting family members. After adjusting for these confounding factors and day of PCR testing by multivariate logistic regression analysis, the VE in the period June–July (delta variant proportion 45%) was 92% and 79% in the period August–September (delta variant proportion 89%). The adjusted VE for homestay, hotel-based isolation and quarantine, and hospitalization was 78%, 77%, and 97%, respectively. Despite declining slightly, VE against hospitalization remained robust for ~3 months after the second dose. Vaccination policymaking will require longer-term monitoring of VE against new variants.

## 1. Introduction

The messenger RNA (mRNA) vaccine against severe acute respiratory syndrome–coronavirus 2 (SARS-CoV-2) was developed rapidly and found to be highly efficacious by clinical trials [1,2]. The efficacy against symptomatic coronavirus disease 2019 (COVID-19) was 95.0% for BNT162b2 (Pfizer Inc., New York, NY, USA; BioNTech Manufacturing GmbH, Mainz, Germany; Comirnaty) [1] and 94.1% for mRNA-1273 (Moderna Inc., Cambridge, UK; MA, USA; Spikevax) [2]. In Israel, where BNT162b2 was authorized for emergency use and administered to the public for the first time, significant effectiveness against COVID-19 was reported after 14 days from the first dose, and the adjusted estimate of vaccine effectiveness (VE) against SARS-CoV-2 infection at ≥7 days after the second dose was 95.3% (95% confidence interval [CI]: 94.9–95.7) [3]. Adjusted estimates of VE against symptomatic COVID-19, COVID-19-related hospitalization, severe or critical COVID-19-related hospitalization, and COVID-19-related death were 91.5% (95% CI: 90.7–92.2%), 97.2% (95% CI: 96.8–97.5%), 97.5% (95% CI: 97.1–97.8%), and 96.7% (95% CI: 96.0–97.3%), respectively [3]. Since that time, real-world evidence of high VE against COVID-19 was reported by epidemiologic studies [4,5]. However, concerns about reduced VE against COVID-19 infection due to reductions in neutralizing activity from waning immunity induced by the mRNA vaccine and the appearance of virus variants have been raised [6,7,8]. On the other hand, VE against serious illnesses requiring hospitalization was reported to still be high not only 3 months after the second vaccination but also 6 months after [9,10,11,12,13,14]. The degree of VE varies depending on the age and sex of the target population, underlying diseases, immune attenuation due to the time since the last vaccination, and the status of the primary epidemic strains; however, studies thoroughly examining these factors in Asian countries such as Japan remain limited [15].

Non-pharmaceutical public health measures other than vaccines and personal health prevention behaviors reportedly prevent COVID-19 [16,17,18]. A meta-analysis revealed that the incidence of COVID-19 is negatively associated with handwashing (relative risk [RR]: 0.47, 95% CI: 0.19–1.12), mask-wearing (RR: 0.47, 95% CI: 0.29–0.75), and physical distancing (RR: 0.75, 95% CI: 0.59–0.95) [17]. Randomized trials of the effectiveness of non-pharmaceutical interventions such as wearing a surgical mask against COVID-19 infection also reported preventive effects [19]. However, to the best of our knowledge, no VE studies have adjusted for personal protective health behaviors and preventive measures. Most real-world studies examining VE against COVID-19 have been test-negative case–control and cohort studies using large databases to obtain information regarding vaccination, laboratory testing, clinical infection, and demographic characteristics [15]. If these databases did not include information regarding personal protective health behaviors, the VEs were not adjusted for those behaviors.

In Japan, mRNA vaccination with BNT162b2 and mRNA-1273 has been promoted since February 2021, with priority given to healthcare workers, the elderly, and high-risk individuals. Vaccination targeting later expanded to include generally healthy individuals and those over 12 years of age, and by the end of 2021, approximately 70% of the population of Japan had been vaccinated with two doses [20]. As of February 2022, a booster vaccination campaign for a third dose was being conducted in response to the rapid spread of a new mutant strain designated omicron [20]. Evaluating VE is extremely difficult in Japan because there is no framework in place to collect information on vaccination history, morbidity, and hospitalization at the individual level for use in research. Only a few case–control studies using a test-negative design conducted in fever outpatient clinics in Japan have been reported [21,22,23,24]. Significant protective effects were reported from 14 days after the first vaccination, and the VE against symptomatic COVID-19 was reported >90% after 14 days from the second vaccination dose [21,22]. The VE against delta (B.1.617.2) variant-related symptomatic COVID-19 was slightly lower, at approximately 80% [22,24]. However, disease severity-specific VE has not been evaluated yet. In addition, as the prevalence and mortality rates of COVID-19 in Japan were lower than those in other countries even before the introduction of the vaccine [25,26], a positive effect of lifestyle and protective health behaviors among the Japanese population has been suggested. However, factors such as protective health behaviors have not been considered in previous studies of VE [21,22,23,24].

The purpose of this study was to evaluate VE in Japan, adjusting for confounding factors such as personal protective health behavior in relation to the duration since the last vaccination and the proportion of delta variants during the period studied.

## 2. Materials and Methods

### 2.1. Study Setting and Participants

We conducted a case–control study to evaluate VE against SARS-CoV-2 infection and COVID-19-related hospitalization in Saga Prefecture and Fukuoka Prefecture, both of which are in southern Japan. Cases included symptomatic or asymptomatic SARS-CoV-2–infected patients (≥16 years of age) who were diagnosed by polymerase chain reaction (PCR) test taken at public health centers as part of an epidemiologic survey for close-contact subjects between 4 June and 26 September 2021. This survey period coincided with the end of the fourth wave of COVID-19, driven by the alpha variant, and the entirety of the fifth wave of COVID-19, driven by the delta (B.1.617.2) variant [27]. According to reports from the public health centers in the study area, the proportion of delta variants during June and July was 45%, while that of August and September was 89%. During the study period in Japan, patients with asymptomatic or minor cases of SARS-CoV-2 infection were advised to stay at home or in a hotel for 10 days, and those requiring treatment were admitted to a hospital for at least 10 days. Patients with mild cases were asked to participate in this study when the public health center reported their PCR results, and hospitalized patients were asked to participate when they were discharged. Controls were selected from individuals with negative SARS-CoV-2 PCR test results who were in close contact with cases and either lived in the same household or had been in face-to-face proximity of within 1 m with a case-patient for 15 min without wearing a face mask.

The sample size needed for the study was calculated as follows: assuming α = 0.05/number of variables (20 items) = 0.0025, β = 0.20, VE = 50–70%, and the proportion of vaccination uptake among controls = 40–60%, the required sample size was 35 to 110 cases and 105 to 330 controls.

Participants provided written informed consent after receiving an explanation of the study purpose and contents and the conditions of cooperation in the study. The study protocol was approved by the Ethics Committee of Saga University (approval nos. R2-39, R3-28).

### 2.2. Data Collection

The following data were obtained using a self-administered questionnaire: sex, date of birth, area of residence, weight, height, and underlying medical conditions (such as respiratory disease, heart disease (including hypertension), kidney disease, liver disease (but excluding fatty liver and chronic hepatitis), diabetes, blood diseases (but excluding iron deficiency anemia), diseases that reduce immune function (including malignancies receiving treatment or palliative care), and others). Participants were also asked about their history of pneumonia, general health condition, smoking history, alcohol drinking history, physical activity, sleep, eating habits, frequency of going out, use of daycare, occupation, education, blood type, family members living together, size of residential space, and whether or not there were any COVID-19-infected people around them. Regarding personal protective health behaviors and measures, we asked whether participants had taken any of the following measures: mask-wearing, washing their hands for at least 20 s upon returning home, use of chlorine- or ethanol-based disinfectants, physical distancing, regular ventilation of the home, avoid eating meals with 5 or more persons, and regularly obtaining COVID-19 information. Vaccinated individuals responded to a questionnaire regarding the vaccine used (Pfizer-BioNTech: BNT162b2 or Moderna: mRNA-1273), lot number, number of doses, and date of vaccination, based on their vaccination certificate. Cases responded to whether they had symptoms of COVID-19 and, if so, the type of symptoms, date and time of symptom onset, duration, and severity. Information about the places of care or stay was obtained from the patient, the city office, and the hospital.

### 2.3. Statistical Analysis

The primary analysis assessed the VE against SARS-CoV-2 infection and hospitalization of at least one dose of either vaccine, two doses, or partial vaccination (only one dose) compared with no vaccination. Subgroup analyses were performed to estimate VE according to duration since the last vaccination dose, severity, and strain-specific protection at the time of diagnosis. We first performed bivariate analyses to assess differences in indicators of background characteristics such as area of residence, date of PCR testing, lifestyle, and personal protective health behaviors between cases and controls using the chi-square test or Wilcoxon rank-sum test. Background characteristic variables that exhibited a *p*-value < 0.0025 (0.05/20) or appeared to be medically related to COVID-19 and vaccination status were considered potential confounders for adjustment. Multivariable logistic regression models were constructed to calculate odds ratios (ORs) with 95% confidence intervals (CIs). We employed the following continuous and categorical variables for adjustment: sex, age (in 10-year intervals), area of residence, underlying medical conditions, date of PCR test, handwashing for 20 s, current alcohol drinking, residence in a single-family home, and family members commute to work or school. Adjusted VE was calculated as (1–adjusted OR) × 100 (%). Commercial software (ver. 9.4 for Windows; SAS Institute, Cary, NC, USA) was used for all statistical analyses.

## 3. Results

During the survey period, we explained the purpose of the study and then mailed questionnaires to 612 potential cases, and a total of 398 (65.0%) responded. A small number of close contacts of the cases tested negative by PCR; thus, 222 subjects became control candidates. We explained the purpose of the study to the potential controls and then mailed each a questionnaire, receiving responses from 179 (80.6%) candidates. In total, we enrolled 398 cases (male: 208, female: 190; mean age: 41.7 ± 14.7 years) and 179 controls (male: 69, female: 110; mean age: 46.9 ± 19.3 years) for this analysis.

Table 1 summarizes the characteristic of the study subjects. There were no significant differences in terms of the date of PCR testing or the presence of underlying diseases. In terms of protective health behaviors and living environment, cases were significantly less likely than controls to spend more than 20 s washing their hands, to live in a single-family home, and to have a family member living with them who commuted to and from work or school.

Table 2 summarizes the vaccination status of the study subjects. The proportion of unvaccinated cases was significantly higher than that of controls, and even among those vaccinated, the largest proportion was less than 13 days after the first dose. There was no significant difference in terms of which vaccine was used. Cases diagnosed in June and July were mostly elderly, whereas those diagnosed in August and September were generally younger (Appendix A).

Table 3 summarizes factors associated with COVID-19 infection. One dose or two doses of COVID-19 vaccine, female sex, washing hands for >20 s each time, living in a single-family home, and having family members who commute to work or school were negatively associated with COVID-19 infection. In contrast, age and alcohol consumption were positively associated with infection. After adjustment for these factors and PCR test date, adjusted ORs for one and two doses of vaccine in June and July, when the proportion of delta variant was 45%, were 0.38 (95% CI: 0.07–2.10) and 0.08 (95% CI: 0.01–0.65), respectively, and adjusted ORs for one and two doses of vaccine in August and September, when the delta variant proportion was 89%, were 0.32 (95% CI: 0.17–0.60) and 0.21 (95% CI: 0.11–0.40), respectively (Table 4). Adjusted ORs for one and two doses of COVID-19 vaccine against COVID-19-related hospitalization were lower than those against COVID-19-related hotel-based isolation and quarantine, or homestay.

Figure 1 shows adjusted values of VE against COVID-19 infection and COVID-19–related hospitalization according to time since vaccination. The adjusted VE values against COVID-19 infection were statistically significant after more than 14 days from the first dose, with peak effectiveness occurring less than 13 days from the second dose. In contrast, the adjusted VE values against COVID-19-related hospitalization were statistically significant even less than 13 days from the first dose and increased overtime after the second dose.

Unlike the effectiveness against COVID-19-related hotel-based isolation and quarantine or homestay, VE against COVID-19-related hospitalization did not decline (Appendix A). We estimated the adjusted VE against COVID-19 during the time in which the delta was predominant (August and September) according to the place of care or stay. The adjusted VE against COVID-19 with homestay or hotel-based isolation for two doses of vaccine was 74% (95% CI: 50–87%), whereas that against hospitalization was 98% (95% CI: 85–99%). There were no significant differences for aOR against COVID-19 for the age group (<60 and ≥60 years old or vaccine manufacturer (Pfizer and Moderna) (Appendix A).

## 4. Discussion

In this case–control study conducted from June to September 2021 in Japan, we found that the overall VE of the mRNA COVID-19 vaccine was 65% (95% CI: 39–80%) for a single dose and 81% (95% CI: 65–89%) for two doses. The adjusted VE of two doses against COVID-19 infection in August and September, when the delta variant was predominant, was 79% (95% CI: 60–89%), which was slightly lower than that in June and July (92%, 95% CI: 35–99%) when the proportion of delta variant was less than 50%. Importantly, the adjusted VE of two doses of mRNA vaccine against hospitalization was robust, at 98% (95% CI: 85–99%), even when the delta variant was predominant (August and September). We also examined the VE against COVID-19 in Japan after adjusting for confounding factors, including personal protective health behaviors.

Both asymptomatic and symptomatic mild cases of COVID-19 were included in this analysis. In contrast, case–control studies conducted in Japan using a test-negative design at clinics for fever outpatients during the same period were limited to symptomatic cases [21,23]. Arashiro et al. reported that the VE against symptomatic COVID-19 in August 2021 in a metropolitan area for two doses of mRNA vaccine was 87% (95% CI: 80–91%) [21]. Maeda et al. reported that the VE during the July to September 2021 period in a nationwide setting was 88.7% (95% CI: 78.8–93.9%) [22]. The overall VE of our study was slightly lower, owing to the inclusion of asymptomatic cases; however, the CIs of the present study overlapped with those of other studies conducted in Japan. Thus, we believe that our VE estimates are comparable, despite the differences in sites, study design, and case definitions.

Observational studies have reported lower VE against delta variant COVID-19 infection compared with the alpha variant, especially among vulnerable groups; however, the VE in those studies was still high in the short term after the second vaccine dose [10,11,12]. A systematic review reported that the VE of two doses of the mRNA vaccine against COVID-19 infection was 74% (95% CI: 62–85%) for the delta variant [13]. Our results are in line with previous reports, with a VE against infection after two doses of any vaccine of 79% (95% CI: 60–89%) during the period in which the delta variant was dominant. Although lower than the VE against infection with the alpha variant, this value met the minimum VE of 50% recommended by the World Health Organization [28]. The VE against severe disease requiring hospitalization was quite high, even when the delta variant was dominant. The adjusted VE against COVID-19-related hospitalization was 98% (95% CI: 85–99%) in our study (Appendix A), which is similar to the pooled VE against severe outcomes after two doses reported in a recent meta-analysis of 98.5% (95% CI: 95–99%) [29]. These results suggest that the mRNA vaccines help prevent severe COVID-19 breakthrough infection. Some recent studies reported a diminished VE against omicron (B1.1.529) variant COVID-19 infection [30,31] due to the potential for immune evasion, which was supported by in vitro neutralization assays [8,32,33,34]. In contrast to the VE against the delta variant, the VE against the omicron variant in terms of hospitalization was reportedly lower, at 65% (95% CI: 51–75%), although it remained above 50% [35]. However, it was reported that the VE against hospitalization improved to 86% (95% CI: 77–91%) after three doses of the vaccine [8,35]. Recent studies on SARS-CoV-2 evolutions have reported the intrinsically high recombination and mutation rate in the Spike genes, which may increase binding affinity to human ACE2 receptors [36,37]. Further studies to monitor VE against the omicron variant, which had the highest number of recurrent mutations [37], are needed to understand the role of prevention measures with vaccination and to develop effective vaccination policies.

Several studies have reported the effectiveness of non-vaccine public health measures and personal health protection behaviors in preventing COVID-19 [16,17,18,19]. A systematic review and meta-analysis reported RRs of COVID-19 infection for handwashing, mask-wearing, and physical distancing of 0.47 (95% CI: 0.19–1.12), 0.47 (95% CI: 0.29–0.75), and 0.75 (95% CI: 0.59–0.95), respectively [17]. In this study, the OR of handwashing for more than 20 s each time against COVID-19 infection was 0.60 (95% CI: 0.41–0.88), which indicates that personal protective health behaviors are negatively associated with COVID-19 infection. People who take precautions such as thorough hand washing are likely to be protected not only from contact infection but also from other routes of infection. Alcohol consumption was more prevalent in the case group, which may reflect more opportunities for drinking or socializing and, thus, more opportunities for infection. In addition, fewer people in the case group lived in a single-family home or had family members who commuted to work or school, reflecting the fact that many of the cases were younger. Few studies using large databases conducted in other countries have adjusted for confounding associated with protective health behaviors and residential environment. One of the advantages of our study is that we were able to evaluate VE after adjusting for these factors.

Other advantages of this study were as follows. First, the definitions of cases and controls were based on PCR test results, resulting in few misclassifications. Second, vaccination history was confirmed by vaccination certificates, thereby minimizing recall bias. In comparison with previous studies in Japan, our study had two more advantages. We recruited the participants from among subjects who underwent PCR testing after coming into close contact with COVID-19 patients, independent of symptoms; thus, the selection was not biased by medical treatment-seeking behavior or test-seeking behavior of the participants. Owing to this recruiting method, we could estimate VE against infection in real-world situations. Finally, we were able to confirm that a high VE was maintained in cases that required hospitalization in Japan.

Our study also had several limitations. First, we did not collect information regarding the SARS-CoV-2 strain in each case. We collected information regarding the epidemic strain in the study area, and we estimated VE by period according to the proportion of delta variants detected in the area. Second, we could not evaluate VE for longer periods after vaccination because the longest duration was 3 months. Third, we did not have clinical information regarding the severity of each case; instead, we used the settings of care or stay (i.e., hospital, hotel, or home) as surrogates for indicators of severity. Not all hospitalized cases were severe; thus, not all cases needed ventilators or extracorporeal membrane oxygenation in Japan. Fourth, cases involving severe disease or death were not included in the study. This selection bias may have led to an underestimation of VE against severe diseases. Fifth, we could not examine differences in VE according to age, presence of high-risk comorbid conditions, or vaccine manufacturer due to the limited sample size. Finally, protective health behaviors were self-reported. In this study, we found that handwashing was negatively associated with COVID-19; however, reverse causation might have occurred due to the retrospective design.

## 5. Conclusions

This case–control study examined the real-world effectiveness of mRNA COVID-19 vaccines among the Japanese population. After adjustment for confounding factors, including protective health behaviors and environmental factors, the VE against COVID-19 infection were 65% (95% CI: 39–80%) for a single dose and 81% (95% CI: 65–89%) for two doses. Although the VE of two doses against infection was slightly lower during the period in which the delta variant was dominant, the VE against hospitalization remained above 90%. Further studies are needed to monitor VE against new variants and the effectiveness of booster doses over time.

## Figures and Tables

**Figure 1 vaccines-10-00779-f001:**
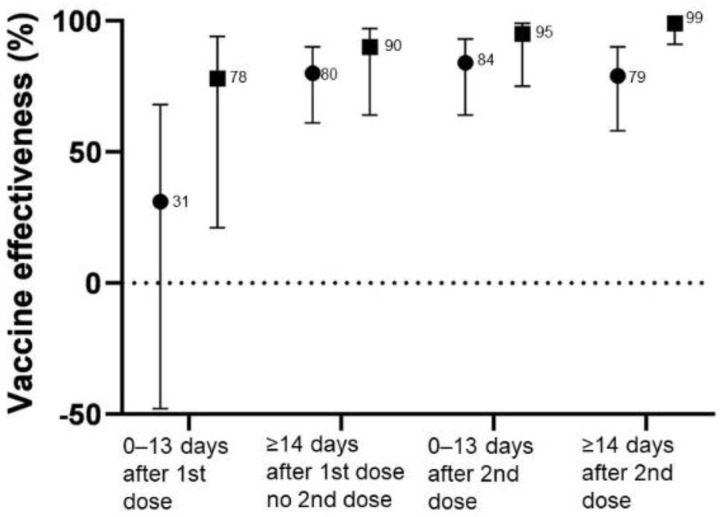
Vaccine effectiveness against any COVID−19 infection (●) and COVID−19related hospitalization (■).

**Table 1 vaccines-10-00779-t001:** Characteristics of the study participants.

		Cases (*n* = 398)		Controls (*n* = 179)		
		*n*	%	*n*	%	*p*-Value *
Sex	Female	190	47.7	110	61.5	0.003
Area	Fukuoka city	381	95.7	158	88.3	<0.001
Age group	16–19	22	5.5	25	14.0	<0.001
(years)	20–29	68	17.1	18	10.1	
	30–39	88	22.1	18	10.1	
	40–49	93	23.4	38	21.2	
	50–59	79	19.8	33	18.4	
	60–69	30	7.5	21	11.7	
	≥70	18	4.5	26	14.5	
PCR test	June	3	0.8	4	2.2	0.47
	July	61	15.3	30	16.8	
	August	318	79.9	138	77.1	
	September	16	4.0	7	3.9	
Any comorbidity		96	24.1	46	25.7	0.68
Protective health behavior						
	Wear a mask during contact with anyone	380	95.5	175	97.8	0.184
	Wash hands for ≥20 s each time	242	60.8	129	72.1	0.009
	Use a hand sanitizer	358	89.9	160	89.4	0.836
	Keep >1.5 m distance during contact with anyone	325	81.7	155	86.6	0.143
	Regular ventilation and disinfection	326	81.9	145	81.0	0.795
	Dining with 5 or more people	22	5.5	9	5.0	0.806
	Obtain information on COVID-19 regularly	327	82.2	148	82.7	0.879
Lifestyle	Current smoking	102	25.6	41	22.9	0.297
	Current alcohol drinking	219	55.0	78	43.6	0.034
Live in a single-family home		137	34.4	78	43.6	0.037
Commute to work or school		315	79.1	135	75.4	0.318
Family members commute to work or school		285.0	71.6	158.0	88.3	<0.001

* The Chi-square test was used to analyze the differences between cases and controls.

**Table 2 vaccines-10-00779-t002:** Vaccination status.

		Cases (*n* = 398)		Controls (*n* = 179)		
		*n*	%	*n*	%	*p*-Value *
Vaccination dose	0	286	71.9	75	41.9	<0.0001
	1	66	16.6	41	22.9	
	2	46	11.6	63	35.2	
Vaccine type	BNT162b2	96	85.7	90	86.5	0.382
	mRNA-1273	16	14.3	14	13.5	
Days after the second dose; the mean (SD)		42.9	(39.4)	27.4	(21.5)	0.02
Unvaccinated		286	71.9	75	41.9	<0.0001
Less than 13 days from 1st dose to PCR test		39	9.8	14	7.8	
More than 14 days from 1st dose to PCR test		27	6.8	27	15.1	
Less than 13 days from 2nd dose to PCR test		16	4.0	23	12.8	
14 to 29 days from 2nd dose to PCR test		5	1.3	19	10.6	
30 to 59 days from 2nd dose to PCR test		12	3.0	13	7.3	
More than 60 days from 2nd dose to PCR test		13	3.3	8	4.5	

SD: standard deviation. * The Chi-square test and t-test were used to analyze the differences between cases and controls.

**Table 3 vaccines-10-00779-t003:** Factors associated with COVID-19 infection in Japan, 4 June–26 September.

		Cases	Controls	Crude OR	95% CI
Vaccination dose	0	286	75	1	(Reference)
	1	66	41	0.42	(0.27–0.67)
	2	46	63	0.19	(0.12–0.30)
Sex	Male	208	69	1	(Reference)
	Female	190	110	0.58	(0.40–0.83)
Area	Fukuoka city	381	158	1	(Reference)
	Saga city	17	21	0.34	(0.17–0.65)
Age group (years)	16–19	22	25	1	(Reference)
	20–29	68	18	4.29	(1.98–9.30)
	30–39	88	18	5.56	(2.59–11.9)
	40–49	93	38	2.78	(1.40–5.52)
	50–59	79	33	2.72	(1.35–5.49)
	60–69	30	21	1.62	(0.73–3.61)
	≥70	18	26	0.79	(0.34–1.81)
PCR test	June	4	3	1	(Reference)
	July	30	61	2.71	(0.57–12.9)
	August	138	318	3.07	(0.68–13.9)
	September	7	16	3.05	(0.54–17.4)
Any comorbidity		96	46	0.92	(0.61–1.38)
Protective health behavior					
	Wear a mask during contact with anyone	380	175	0.48	(0.16–1.45)
	Wash hands for over 20 s each time	242	129	0.60	(0.41–0.88)
	Use a hand sanitizer	358	160	1.06	(0.60–1.89)
	Keep >1.5 m distance during contact with anyone	325	155	0.69	(0.42–1.14)
	Regular ventilation and disinfection	326	145	1.06	(0.38–1.67)
	Dining with 5 or more people	22	9	1.11	(0.50–2.45)
	Obtain information on COVID-19 regularly	327	148	0.97	(0.61–1.54)
Lifestyle	Current smoking	102	41	1.16	(0.77–1.76)
	Current alcohol drinking	219	78	1.58	(1.11–2.25)
Live in a single-family home		137	78	0.68	(0.48–0.98)
Commute to work or school		315	135	1.24	(0.82–1.88)
Family members commute to work or school		285	158	0.34	(0.21–0.57)

**Table 4 vaccines-10-00779-t004:** Adjusted odds ratio (aOR) against COVID-19 according to diagnosis period and place of care or stay.

		Total	Diagnosis Period		Place		
			June and July	August and September	Hospital	Hotel	Home
		aOR * (95%CI)	aOR * (95%CI)	aOR * (95%CI)	aOR * (95%CI)	aOR * (95%CI)	aOR * (95%CI)
Vaccination dose	0	1	1	1	1	1	1
		(Reference)	(Reference)	(Reference)	(Reference)	(Reference)	(Reference)
	1	0.35	0.38	0.32	0.15	0.29	0.36
		(0.20–0.61)	(0.07–2.10)	(0.17–0.60)	(0.05–0.41)	(0.12–0.68)	(0.19–0.71)
	2	0.19	0.08	0.21	0.03	0.23	0.22
		(0.11–0.35)	(0.01–0.65)	(0.11–0.40)	(0.01–0.13)	(0.08–0.60)	(0.11–0.45)
Unvaccinated		1	1	1	1	1	1
		(Reference)		(Reference)	(Reference)	(Reference)	(Reference)
Less than 13 days from 1st dose to PCR test		0.69	1.21	0.68	0.22	0.76	0.79
		(0.32–1.48)	(0.13–11.2)	(0.27–1.69)	(0.06–0.79)	(0.24–2.22)	(0.31–2.05)
More than 14 days from 1st dose to PCR test		0.20	0.09	0.18	0.10	0.11	0.21
		(0.10–0.39)	(0.01–0.94)	(0.08–0.39)	(0.03–0.36)	(0.03–0.38)	(0.09–0.48)
Less than 13 days from 2nd dose to PCR test		0.16	0.69	0.12	0.05	0.14	0.16
		(0.07–0.36)	(0.03–17.7)	(0.05–0.29)	(0.01–0.25)	(0.04–0.58)	(0.06–0.46)
More than 14 days from 2nd dose to PCR test		0.21	0.01	0.30	0.01	0.31	0.25
		(0.10–0.42)	(<0.001–0.15)	(0.14–0.65)	(<0.001–0.09)	(0.10–1.02)	(0.11–0.57)

* Sex, age (10-year intervals), area of residence, underlying medical conditions, PCR test date, handwashing for 20 s, current alcohol drinking, living in a single-family home, and family members’ commute to work or school.

## Data Availability

The data presented in this study are available on request from the corresponding author (M.H.). The data are not publicly available due to privacy concerns.

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
