# Peer review of "Real-World Effectiveness of the mRNA COVID-19 Vaccines in Japan: A Case–Control Study"

_vaccines, 2022, doi:10.3390/vaccines10050779_

Round 1
Reviewer 1 Report
The authors conducted a study to evaluate the vaccine effectiveness of the two mRNA vaccines in two regions of southern Japan, during the summer of 2021, after adjusting for various confounding factors that are associated with protective health behaviors and residential environment. Importantly, the authors recruited the participants from among subjects who underwent PCR testing after coming into close contact with COVID-19 patients, independent of symptoms. This is a very important aspect.
The results are very interesting, make sense and are in accordance with the available literature on this subject. Overall, they are a nice addition to the current knowledge and they may help to guide future policies, when new variants emerge and new vaccines are produced. The manuscript was very well written, easy to follow, with very few and minor mistakes. In addition, the authors included a very good set of references. The discussion was very good and the authors clearly presented the advantages and limitations of their study. The number of cases and controls that the authors used in this study for testing confounding factors is not very high, but obtaining such kind of data is not an easy task anyway. I believe that it will work as a starting point for continuing their study in the months and maybe even in the years ahead.
Some minor comments that could benefit the manuscript:
First paragraph of introduction: It would be useful to include here the reference of Cameroni et al. (doi: 10.1038/s41586-021-04386-2) or somewhere in the discussion and incorporate its findings in the discussion.
Also, it would be useful to further include somewhere in the introduction or discussion a sentence or two about the intrinsically high recombination and mutation rate that the Spike demonstrates, especially at the variants of concern (see doi: 10.1093/molbev/msab292, and doi: 10.3390/v14040707). It is important for the reader to understand that SARS-CoV-2 evolves very fast and that the current mRNA vaccines are based on a highly unstable (from an evolutionary perspective) region. Thus, understanding the role of prevention measures together with vaccination is very important.
Concerning which strains were circulating during that period in Japan, maybe the Nextstrain/GISAID databases could be useful (doi:10.2807/1560-7917.ES.2017.22.13.30494, doi:10.1093/bioinformatics/bty407).
https://nextstrain.org/ncov/gisaid/global
In this database, the authors can select the time period and the country and visualize which strains were circulating.
Line 31: vaccines
Line 32: of which vaccines exactly
Line 274: in vitro: please italicize
Sentence 272-274: Omicron twice
Line 290: is it more opportunities for drinking or socializing?
Author Response
We thank Reviewer #1 for his/her constructive comments on our manuscript. We respectfully respond to his/her concerns as follows:
Reviewer 1
Some minor comments that could benefit the manuscript:
First paragraph of introduction: It would be useful to include here the reference of Cameroni et al. (doi: 10.1038/s41586-021-04386-2) or somewhere in the discussion and incorporate its findings in the discussion.
Response: We thank Reviewer #1 for his/her precious suggestion. We added this articles in the first paragraph of introduction (line 66) and discussion (line 274 and 278).
Also, it would be useful to further include somewhere in the introduction or discussion a sentence or two about the intrinsically high recombination and mutation rate that the Spike demonstrates, especially at the variants of concern (see doi: 10.1093/molbev/msab292, and doi: 10.3390/v14040707). It is important for the reader to understand that SARS-CoV-2 evolves very fast and that the current mRNA vaccines are based on a highly unstable (from an evolutionary perspective) region. Thus, understanding the role of prevention measures together with vaccination is very important.
Response: We thank Reviewer #1 for his/her precious suggestions. We added some sentencec with reference to these articles in the third paragraph of discussion (line 288-293).
Concerning which strains were circulating during that period in Japan, maybe the Nextstrain/GISAID databases could be useful (doi:10.2807/1560-7917.ES.2017.22.13.30494, doi:10.1093/bioinformatics/bty407).https://nextstrain.org/ncov/gisaid/global
In this database, the authors can select the time period and the country and visualize which strains were circulating.
Response: We appreciate this valuable information. We checked strains circulating during the study period in Japan with Nextstrain/GISAID. We found that proportions of 21J delta and 21I delta during June and July were 30 to 40 % and 4 to 10 %, respectively. While those proportions during August and September were 70 to 80 % and 10 to 15%, respectively. We added a reference of GISAID databases in the study setting (Line 118). In addition, we had information on circulating strains during the study period in the study region from the public health center, so we added sentences regarding the exact proportion of circulating strain in the study setting (Line 119-120).
Line 31: vaccines
Response: We corrected it (Line 31).
Line 32: of which vaccines exactly
Response: We added the vaccine type in Line 33.
Line 274: in vitro: please italicize
Response: We corrected it (Line 283).
Sentence 272-274: Omicron twice
Response: We deleted some words (Line 274).
Line 290: is it more opportunities for drinking or socializing?
Response: We added ‘socializing’ (Line 295).
Reviewer 2 Report
This paper reports a timely and well-done case control study. It follows research design and statistical protocols carefully. Prior to publication, the paper should be checked one more time for accuracy and grammar.
Author Response
We appreciate for reviewing our manuscript. According to Reviewer #2 comments, we checked one more time for accuracy and grammar.